# Association of Socio-Demographic and Climatic Factors with the Duration of Hospital Stay of Under-Five Children with Severe Pneumonia in Urban Bangladesh: An Observational Study

**DOI:** 10.3390/children8111036

**Published:** 2021-11-11

**Authors:** K. A. T. M. Ehsanul Huq, Michiko Moriyama, Ryota Matsuyama, Md Moshiur Rahman, Reo Kawano, Mohammod Jobayer Chisti, Md Tariqujjaman, Nur Haque Alam

**Affiliations:** 1Graduate School of Biomedical and Health Sciences, Hiroshima University, Kasumi 1-2-3, Minami-ku, Hiroshima 734-8553, Japan; morimich@hiroshima-u.ac.jp (M.M.); rmatsuyama@hiroshima-u.ac.jp (R.M.); moshiur@hiroshima-u.ac.jp (M.M.R.); 2Clinical Research Center in Hiroshima, Hiroshima University Hospital, Hiroshima 734-8551, Japan; rkawano@hiroshima-u.ac.jp; 3International Centre for Diarrhoeal Disease Research, Dhaka 1212, Bangladesh; chisti@icddrb.org (M.J.C.); md.tariqujjaman@icddrb.org (M.T.); nhalam@icddrb.org (N.H.A.)

**Keywords:** severe pneumonia, under-five children, length of hospital stay, climate factors, Bangladesh

## Abstract

Severe pneumonia is one of the leading contributors to morbidity and deaths among hospitalized under-five children. We aimed to assess the association of the socio-demographic characteristics of the patients and the climatic factors with the length of hospital stay (LoS) of under-five children with severe pneumonia managed at urban hospitals in Bangladesh. We extracted relevant data from a clinical trial, as well as collecting data on daily temperature, humidity, and rainfall from the Meteorological Department of Bangladesh for the entire study period (February 2016 to February 2019). We analyzed the data of 944 children with a generalized linear model using gamma distribution. The average duration of the hospitalization of the children was 5.4 ± 2.4 days. In the multivariate analysis using adjusted estimation of duration (beta; β), extended LoS showed remarkably positive associations regarding three variables: the number of household family members (β: 1.020, 95% confidence intervals (CI): 1.005–1.036, *p* = 0.010), humidity variation (β: 1.040, 95% Cl: 1.029–1.052, *p* < 0.001), and rainfall variation (β: 1.014, 95% Cl: 1.008–1.019), *p* < 0.001). There was also a significant negative association with LoS for children’s age (β: 0.996, 95% Cl: 0.994–0.999, *p* = 0.006), well-nourishment (β: 0.936, 95% Cl: 0.881–0.994, *p* = 0.031), and average rainfall (β: 0.980, 95% Cl: 0.973–0.987, *p* < 0.001). The results suggest that the LoS of children admitted to the urban hospitals of Bangladesh with severe pneumonia is associated with certain socio-demographic characteristics of patients, and the average rainfall with variation in humidity and rainfall.

## 1. Introduction

Pneumonia is a leading cause of death accounting for about 15% of all deaths under-five years old, particularly in South Asia and sub-Saharan Africa in 2017 [1]. Several socio-demographic factors, such as age and gender, have been shown to be associated with the severity of pneumonia [2,3]. Children aged one month to five years old, and being male, are more likely to present with pneumonia [2,3]. It has also been reported that a lack of maternal education and lower family income are significantly related to the increase in severity of childhood pneumonia [4]. Female children younger than 12 months old and who were malnourished were found to be associated with childhood pneumonia-related death [5]. Moreover, pneumonia in malnourished children has a greater risk of death, compared to pneumonia alone, worldwide [6]. One study observed that children who were preterm and low birthweight had a higher risk of developing severe pneumonia and requiring admission to intensive care units (ICU) when compared to full term and normal weighted babies [7]. Another study in the UK also revealed that infants born by caesarean section had an increased risk of having severe pneumonia [8]. A study in China noted higher deaths from pneumonia in rural areas compared to urban areas [9].

Climate change impacts the health of children regarding all types of infectious diseases, including pneumonia [10]. Bangladesh is directly affected by global warming and has ranked nineth among the topmost 10 countries that are globally affected by the change [11]. The weather pattern of this country has been changing, with temperatures increasing between 0.6 and 2.0 degrees Celsius over the last 100 years [12]. Bangladesh has three distinct seasons: summer between March and June with temperatures from 30 degrees Celsius (°C) to 40 °C, rainy between June and October, and winter between October and March, with an average temperature of about 10 °C. April is the warmest month, with a peak temperature of 40 °C, and January is the coldest month, with a drop in temperature to an average of 10 °C throughout the country [13]. A study in rural Bangladesh has reported a positive association between the temperature and humidity and the length of stay (LoS) of children hospitalized for severe pneumonia [14]. The duration of the hospital stay is usually indicative of the severity of disease [15], and its rise causes a significant economic burden in both developed and developing countries [16]. The peak of daily temperature, humidity, and rainfall increases the risk of childhood pneumonia [2,10]. Therefore, climate variability can affect the severity of childhood pneumonia and might influence the length of the hospital stay.

Children with pneumonia are recommended to be managed at primary health care facilities and referred to higher-level health care facilities for treatment when their condition becomes severe [17,18]. Unfortunately, 22% of such referred children in Bangladesh are not admitted to those health care facilities due to a shortage of beds, leading to their heightened risk of morbidity and death [19]. An inadequate number of beds is a constraint for the hospitalized management of children with severe pneumonia because consequently some are not able to be admitted and are deprived of getting appropriate treatment from the hospitals [19]. A longer LoS in the hospital increases the occupancy of the beds and the total cost of hospitalization [20,21].

There has been no study in Bangladesh that has assessed the correlation of the socio-demographic and climate factors with the LoS of under-five children with severe pneumonia in urban sites. Identifying such correlating factors could enable the targeting of measures to reduce LoS, which would in turn result in the availability of hospital treatment to more children suffering from severe pneumonia. For example, if high environmental temperature and humidity were found to influence the patients’ LoS in hospital, the provision of effective air conditioning systems in hospitals along with an uninterrupted power supply might act to reduce the LoS.

Our study, which was nested onto a clinical trial, was designed to assess the association between factors related to the parents’ socio-demographic characteristics, children’s birth history (normal, preterm, and post-term; normal vaginal, caesarean section), and nutritional status, and variations in climatic factors (temperature, relative humidity, and rainfall) with the LoS of under-five year old children hospitalized for severe pneumonia in urban areas of Bangladesh.

## 2. Materials and Methods

### 2.1. Study Design

This was an observational study design following the Strengthening the Reporting of Observational studies in Epidemiology (STROBE) checklist for reporting.

### 2.2. Data Collection

We extracted data from two different resources. The clinical data were collected from a cluster randomized control trial (ClinicalTrials.gov, Identifier: NCT02669654) conducted among the under-five children with severe pneumonia. Children who were included in the clinical trial were admitted to different pediatric departments of general hospitals (i.e., Dhaka Shishu Hospital, Dhaka Medical College and Hospital, icddr,b Dhaka Hospital, and other nearby hospitals) in the study areas for the management of severe pneumonia. That study enrolled 954 admitted children with severe pneumonia from those hospitals, and among them 946 children completed the study. We analyzed the data of 944 children for this study, excluding two children due to missing data (Figure 1). The clinical trial was conducted between February 2016 and February 2019.

We used the hospitals’ data during the period of hospitalization of the children and computed the LoS in days. For our analysis, we considered individual parents’ background socio-demographic characteristics; birth (full-term, preterm, post-term), delivery history (normal vaginal, cesarean section) of the children, and their nutritional status. We collected data on the climatic factors (temperature, relative humidity, and rainfall) of the study period from the Meteorological Department of Bangladesh (Agargaon, Dhaka 1207, Bangladesh) [22]. Then, from the climate data, we extracted and used the temperature, humidity, and rainfall data for only those days corresponding to the dates of each child’s stay in hospital. The meteorological station called ‘Dhaka Station’ was located at Agargaon, 1.2 km from DSH, 6.2 km from icddr,b Dhaka Hospital, and about 6.3 km radius from other hospitals. We collected daily minimum and maximum values of the factors and computed the average for each day.

### 2.3. Operational Definitions

Pneumonia is identified as rapid breathing (≥50 breaths/min for 2–11 months old and ≥40 breaths/min for 12–59 months old), and/or chest indrawing [23].

Severe pneumonia is described as the appearance of cough or difficulty in breathing and the presence of one or more of the following signs: central cyanosis, hypoxemia (oxygen saturation < 90% on pulse oximetry), severe respiratory distress (grunting, very severe chest indrawing), or pneumonia with danger signs (inability to breastfeed or drink, lethargy, unconsciousness, and/or convulsions) [23].

We defined Z-score <−2 (weight-for-age; ZWA) as malnourished children and Z-score ≥−2 (weight-for-age; ZWA) as without malnutrition [24].

Average temperature, humidity, and rainfall: the average temperature, humidity, and rainfall were calculated as the mean value during the days of hospital stay period of the under-five children with severe pneumonia [25].

Temperature, humidity, and rainfall variation: the magnitudes vary over time during the days of hospital stay. The variation was the dispersion of values from the mean [25].

Daily temperature, humidity, and rainfall range: the daily range was the difference between the maximum and minimum values during the days of the hospital stay period [25].

### 2.4. Study Population

Primary health care in urban areas was provided by the Ministry of Local Government, Rural Development, and Cooperatives (LGRD) of Bangladesh through partnership with non-government organizations. Maternal and child health including respiratory tract infection was treated by the Primary Health Care Centers (PHCCs) [26]. Each PHCC provided primary health care at the urban community levels for populations of 30,000 to 50,000 [27]. Participants of the clinical trial were identified from the eight selected PHCCs of the urban areas of Dhaka city within the existing health systems. Community health workers (CHWs), who resided at the community level, routinely visited door-to-door of his/her catchment areas to find out if there were children with any illness, including pneumonia. If the patients had the symptoms of pneumonia, the health workers sent the patients to the nearby PHCC. Pneumonia children were treated by the PHCC’s physicians, staying and taking medication at their homes. If the children’s condition deteriorated and they developed severe pneumonia, they were transferred from these PHCCs to the referral hospitals. Children with severe pneumonia also were self-referred by their parents to attend the hospitals directly. After admission to the hospitals, the designated study nurse obtained written informed consent from the attending parents or their caregivers for enrollment into the clinical trial and the use of their data for this study. Children with severe pneumonia, aged 2 to 59 months, of both sexes, living in the designated study areas, with written informed consent by their parents or caregivers were included, and children who had severe acute malnutrition (SAM), weight-for-height Z-score [ZWH] <−3, or bilateral pitting edema, or a mid-upper arm circumference [MUAC] of <115 mm), occurring alone or in combination, were excluded from the study, as they need special care (e.g., nasogastric tube feeding, micronutrients, rehabilitation, etc.). The trial collected relevant data of individual children onto a predesigned and pretested “Case Report Form (CRF),” and transcribed them into a database.

### 2.5. Statistical Analysis

Demographic categorical data were presented as frequencies, and continuous data were presented as mean and standard deviation along with their range (maximum and minimum). All relevant study data were analyzed by using SPSS for Windows (version 25.0; SPSS Inc., Chicago, IL, USA) and Epi Info (version 7.0, USD, Stone Mountain, GA, USA). A probability of less than 5% (*p* < 0.05) was considered statistically significant. The strength of associations between LoS (i.e., the response variable in the present study) and other variables was determined by calculating the beta (β) coefficients of bivariate and multivariate generalized linear models that assume gamma distributed error (gamma GLM) and their 95% confidence intervals (CI). As explanatory variables in gamma GLM, we included four categorical variables on children and their family members’ status (children’s sex, nutritional status, birth history, and parents’ occupation), six continuous variables regarding children and their family members’ socio-demographic characteristics (children’s age, the number of family members, the number of siblings, parents’ age, years of parents’ education, and the household income), and six continuous variables on climate (the average temperature, humidity, and rainfall as well as their changes (variation) and ranges). We used the gamma GLM because the response variable data type was as continuous as the LoS in the hospital (time to event), not normally distributed, and skewed to the right. To see the monthly trends of climate variables during the hospitalization of children, a time series analysis was performed, and association was estimated by Pearson’s correlation method. We calculated the average of temperature, humidity, and rainfall by adding the total value divided by the length of the hospitalization period for each child. We also calculated the variation and range of temperature, humidity, and rainfall by using the data for the period of the stay in the hospitals. The range was calculated by subtracting the minimum value from the maximum value of climate factors, and the variation used in this research was the standard deviation of climate factors. We analyzed the explanatory variables and dropped out variables that had variance inflation factor (VIF) values more than ten, due to the presence of multicollinearity.

## 3. Results

### 3.1. Demographic Characteristics of the Participants

Of a total 1693 screened under-five children with severe pneumonia, 954 (56.3%) children who were admitted in different hospitals were enrolled, and of them, the study was completed for 946 (99.2%). We analyzed the data of 944 children for this study (Figure 1). Table 1 shows the demographic characteristics, parents’ occupation, nutritional status and birth history. Fathers of the children were 98.5% employed in the following categories: skilled occupations included office executive, big business, and government officials were 321 (34.1%); unskilled workers such as office nonexecutive, rickshaw, and pushcart puller, and taxi, bus, and tempo (local transport) drivers were 605 (64.4%). Mothers of 871 (92.3%) children were unemployed, including housewives and students. Children of 23.3% were malnourished and 49.0% were delivered by caesarean section. Table 2. expressed other demographic characteristics, the LoS, and the climate (temperature, humidity, and rainfall) data for all enrolled study children and all their corresponding hospital days. The average LoS was 5.4 ± 2.4 days with a range of between 1 day and 16 days.

### 3.2. Days of Hospitalization for Study Children (%)

Children with severe pneumonia were admitted and stayed in hospital between 1 day and 16 days. The highest percentage of children stayed in hospital for 5 days (19.1%) followed by 4 (18.1%), 6 (17.8%), and 3 (12.1%) days (Figure 2), respectively.

### 3.3. Trends of Temperature, Humidity, and Rainfall during Hospitalization

The monthly trend of temperature, humidity, and rainfall during the hospitalization among under-five children with severe pneumonia is shown in Figure 3. We observed there was no statistically significant association of monthly patient stays in the hospital with monthly average temperature (r = 0.114, *p* = 0.500), monthly tem-perature variation (r = −0.170, *p* = 0.315), monthly average humidity (r = 0.108, *p* = 0.525), monthly humidity variation (r = −0.019, *p* = 0.910), monthly average rainfall (r = 0.090, *p* = 0.596), and monthly rainfall variation (r = 0.113, *p* = 0.505). There were downward trends observed for all climate variables between October and February in every year. From April to October in every year, the temperature and humidity were maintained higher. Although rainfall was less in the winter season from December to February in every year, the influence on monthly patient stays in the hospital was not clearly observed.

### 3.4. Results of Generalized Linear Model with Gamma Distribution

We analyzed study data and dropped out the daily range of temperature, humidity, and rainfall variables due to the presence of multicollinearity. Table 3 shows the coefficients (β) with 95% confidence intervals (Cl) and *p*-values of the socio-demographic and climatic factors with respect to the LoS in days that were estimated in the bivariate and multivariate analyses in gamma GLM. After adjusting for the children’s sex, the mother’s and father’s age, the mother’s and father’s education, the mother’s and father’s occupation, the number of siblings, the household income, the birth and delivery history, the children’s age (β: 0.996, 95% Cl: 0.994–0.999, *p* = 0.006), and the well-nutritional status (β: 0.936, 95% Cl: 0.881–0.994, *p* = 0.031), number of household family members (β: 1.020, 95% Cl: 1.005–1.036, *p* = 0.010), the humidity variation (β: 1.040, 95% Cl: 1.029–1.052, *p* < 0.001), average rainfall (β: 0.980, 95% Cl: 0.973–0.987, *p* < 0.001), and rainfall variation (β: 1.014, 95% Cl: 1.008–1.019, *p* < 0.001) were significantly associated with the LoS among under-five children with severe pneumonia.

In summary, our analysis indicates that for every 1% increase in daily humidity variation and 1 mm greater daily rainfall variation, the LoS increased by 4.0% and 1.4%, respectively. On the other hand, with a 1 mm decrease in average rainfall, the LoS increased by 2.0%. The LoS was positively associated with humidity variation and rainfall variation (β: 1.040, 95% Cl: 1.029–1.052, *p* < 0.001 and β: 1.014, 95% Cl: 1.008–1.019, *p* < 0.001), respectively, and negatively associated with average rainfall (β: 0.980, 95% Cl: 0.973–0.987, *p* < 0.001). The results suggested that the greater variations in the ambient humidity and rainfall, and lesser average rainfall, were associated with a longer hospital stay of the children. However, the average temperature and humidity were not associated with the LoS. There was also no significant association between the sex of the children, their parental education, the mother’s occupation, or the household income with the LoS.

## 4. Discussion

Our study demonstrated an association between the LoS of under-five children admitted to urban hospitals with severe pneumonia and their socio-demographic characteristics and some of the climatic factors. Malnourished children in the lower age group and having more household family members had more risk of a longer LoS in the hospital. Humidity and rainfall variation, and reduced average rainfall, also increased the risk of a longer LoS in the hospital.

We observed a significant negative association between the LoS and the age and malnutrition of the children, where the lower in these factors was associated with longer hospital stay of the children. Our study children had a wide distribution of age, 1–59 months, where the lower age of this range was associated with the longest duration of hospitalization. Our study finding is consistent to another study in which infants (less than one-year old) had longer stays than the children of one year and above [28].

Several previous studies in developing countries reported male children receiving better treatment compared to their female counterparts; however, we did not find its impact on LoS—a finding similar to other studies conducted in rural Bangladesh [12], the UK [29], and Iran [30].

We did not observe any association between the occupation of the mother or the father with a longer LoS. However, one study in the UK reported a longer hospitalization of children who had fathers with lower income. Thus, the findings may actually be related to the functional differences in the operations, and therefore outcomes of the health services of individual hospitals, regions, and countries [29].

We noted that there was a positive association between the number of household family members and a longer LoS in the hospital. It was observed that in some families, there was lack of care required to be given to the sick child due to the competing care needs of other family members. The severity of pneumonia may be related to the delay in seeking of care, and thus with a longer stay in those who had more household family members. Some physicians, for cultural reasons, extend their patients’ hospital stay for longer, for fear that the parents cannot take proper care at home due to caring for other family members [31].

With respect to the nutritional status of our study children, we found a negative association of malnutrition with longer LoS in hospital. Similar findings were observed in previous studies, which reported a stronger association between better nutritional status and a shorter LoS in hospital [14,28,32]. One study in China reported that preterm and low birth weight children became more severe and needed more rigorous care compared to normal birth children [14]; however, we did not find any association.

We did not find a significant association of the socio-demographic factors of children’s sex, their birth or delivery history, the parental education and occupation, the household income, or the number of siblings with children’s LoS.

Climatic factors, including the daily humidity and daily rainfall variations, had a significant positive association with the LoS in hospital, with the daily humidity variation having the highest impact. We also observed that greater variations in humidity and rainfall could be associated with a longer hospital stay; the opposite findings have been reported elsewhere [33,34]. In our study, we found average rainfall had a significantly negative association with the LoS in hospital. However, we did not observe any significant association between the average temperature and the average humidity with a longer LoS. Previous studies have attempted to evaluate the relationship between climate variability and hospitalization [35]; however, a single study conducted in rural Bangladesh explored the association between temperature and humidity with the LoS during the hospitalization of under-five children who were suffering from severe pneumonia [14], signifying a knowledge gap for the impact of rainfall on the LoS in hospitals. We observed the trend of humidity variation increasing in summer, average rainfall and rainfall variation in the rainy season, with increasing number of monthly patients’ stay in the hospital. Similar findings were observed in other studies [2,10]. We assessed variations in the climatic factors, particularly the ambient temperature, humidity, and rainfall with the LoS of under-five children with severe pneumonia. In our study hospitals, there were no indoor air controlling facilities; therefore, the ambient temperature and humidity persisted inside the hospitals during the stay of the participants.

Many attributes of children, such as their physical condition, comorbidities, history of previous hospitalization, attitude, and characteristics during their illness, medical practice, hospital admission criteria, treatment cost, and health insurance have been reported to influence the LoS [30,36]. Our study findings might have been influenced by not considering those confounding factors in respect of the LoS in the hospitals due to lack of data. All the study children were diagnosed and managed by qualified physicians in the hospitals according to the WHO guidelines for clinical diagnosis and treatment of pneumonia in under-five children. An earlier study reported that shorter LoS were associated with the management of hospitalized children by more qualified physicians [37].

We conducted our study at urban hospitals, and the mean LoS was longer than a previous study’s findings shown at a rural hospital of Bangladesh [25]. The reason might be that children with more severe conditions were admitted to urban hospitals compared to rural areas. In urban areas, there were more opportunities to consult with a physician or a child specialist, and the children were likely admitted to hospitals only when their condition was more critical. Children were also referred to urban hospitals from the rural ones for better treatment. Rural hospitals have less laboratory investigation facilities, which might cause delay in diagnosis and could be a cause of shorter stay in the hospitals [38]. Similar outcomes were observed at other urban hospitals where the air pollution was more likely to be observed compared to rural areas [39]. However, we did not consider the effects of air pollution on the LoS in the hospitals for our study.

### Strengths and Limitations of This Study

We explored the impact of major climatic factors, namely the ambient temperature, humidity, and rainfall on the duration of the hospital stay. Our study children were managed by qualified physicians according to the severity of their conditions and following the WHO guidelines, thus outcome bias was unlikely. The study children were discharged from the hospitals after recovery from their severe conditions. Our study was conducted at hospitals that provide free care and treatment, and thus the LoS was unlikely influenced by the financial affordability.

There are some limitations in our study. First, our analysis excluded severely malnourished under-five children, and thus the findings may not be generalized for all under-five children. Second, we did not take into account the children’s general health conditions, including comorbidities, which might have influenced the outcome. Third, we did not have data on the time course of the pneumonia before hospitalization; therefore, we could not analyze the association between climate factors and pneumonia before children were admitted into the hospital. Fourth, as we used weather station data in our analysis, the weather variables of humidity and rainfall were not measured in the actual hospital environment; thus, the effects on the LoS need to be interpreted with caution. Fifth, our study children received treatment from different hospitals, and the different rates of patient’s admission flow of those hospitals might have influenced the LoS. However, the study physicians and nurses handled this issue to minimize the influence of timing of patient’s hospital discharge. Finally, as the hospital treatment in our study was free of cost, we did not perform a cost analysis.

## 5. Conclusions

Our study observed patients’ socio-demographic characteristics and some of the climatic factors to be associated with the duration of hospital stay of under-five children admitted to urban hospitals in Bangladesh with severe pneumonia. As Bangladesh is a climate-susceptible country, mass awareness and health education are needed for parents/caregivers and health staff. The changing climate conditions need to be considered with seasonal variation in future health-related policy making, including hospital management. Heating, ventilation, and air conditioning systems can be included in infection control strategies. On the basis of our study findings, the variation of humidity and rainfall issue could be adopted as one of the mitigation strategies, such as air conditioning in order to shorten the LoS and prevent the delay in severe pneumonia recovery time at the hospital, especially for such resource-impoverished countries. Further prospective studies could be conducted among hospitalized children with severe pneumonia in some other countries with similar socio-demographic and climatic factors, where the actual humidity and rainfall measurements are taken inside hospitals so to limit the confounding effects, and for external validation.

## Figures and Tables

**Figure 1 children-08-01036-f001:**
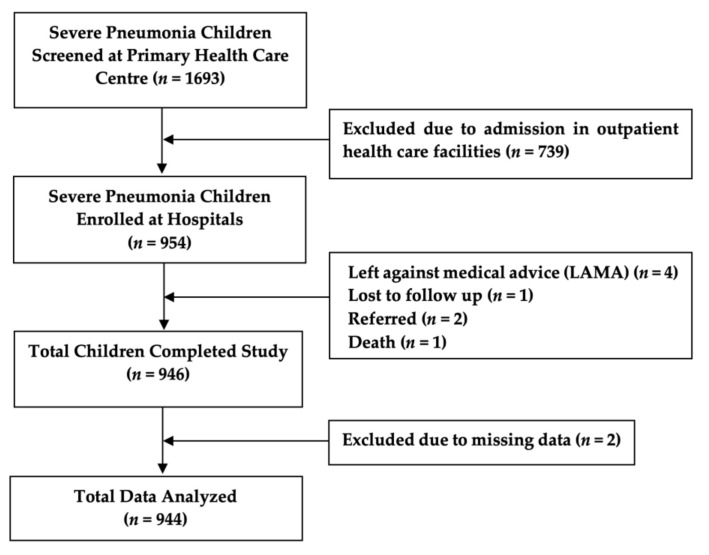
Flow chart of study participants.

**Figure 2 children-08-01036-f002:**
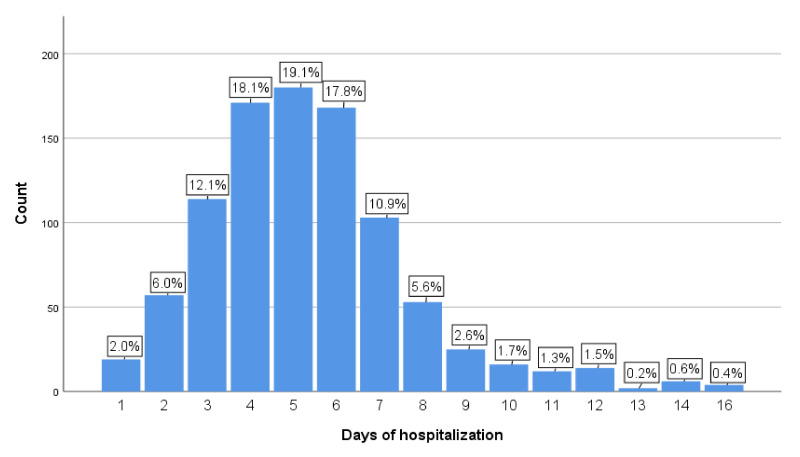
Frequency and percentage of the LoS in hospital in 944 children. LoS in days is shown as horizontal axis and frequency (%) as the vertical axis.

**Figure 3 children-08-01036-f003:**
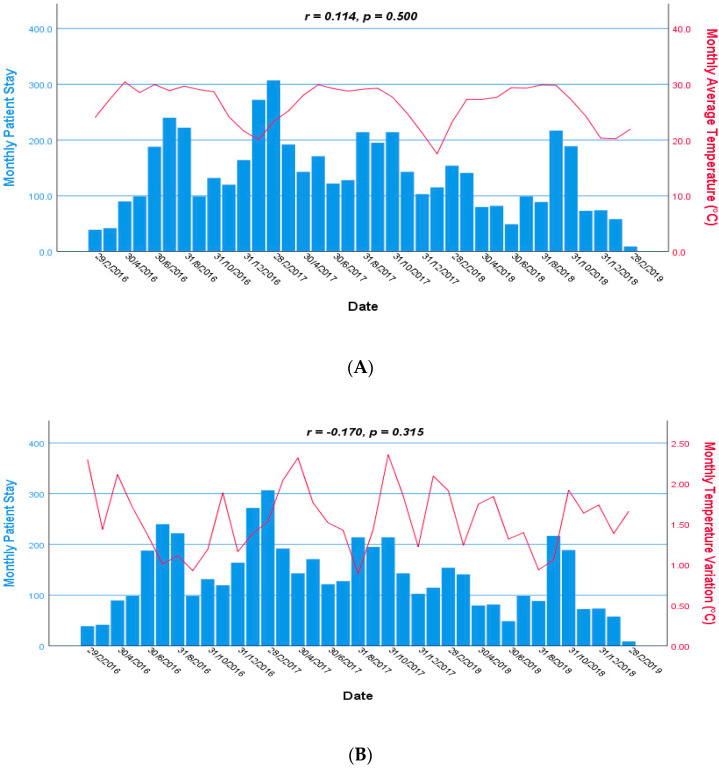
Monthly time series: average temperature (**A**), temperature variation (**B**), average humidity (**C**), humidity variation (**D**), average rainfall (**E**), and rainfall variation (**F**) during the hospital stay of the children with severe pneumonia from 2016 to 2019. Scale of monthly patient stay in the hospital and climate variables are shown in the left and right axis, respectively.

**Table 1 children-08-01036-t001:** Distribution of sex, parents’ occupation, nutritional status, birth and delivery history.

Variables	*n* = 944	%
Children’s sex Male Female	615 329	65.1 34.9
Mother’s Occupation Unemployed (housewife, student) Employed (formal and informal)	871 73	92.3 7.7
Father’s occupation (*n* = 940) Unemployed Unskilled Skilled	14 605 321	1.5 64.4 34.1
Nutritional status Malnutrition, <−2 Z-score (ZWA) Well-nutrition, ≥−2 Z-score (ZWA)	220 724	23.3 76.7
Birth history Full-term Preterm Post-term	849 76 19	89.9 8.1 2.0
Delivery Normal vaginal Cesarean section	481 463	51.0 49.0

**Table 2 children-08-01036-t002:** Mean (SD) and range of other continuous demographic variables.

Variables	Mean ± SD	Min–Max
Age (months)	12.0 ± 10.2	2.0–57.1
Mother’s age (years)	24.6 ± 4.9	12.0–42.0
Father’s age (years)	31.8 ± 6.3	19.0–70.0
Mother’s education (years of schooling)	7.3 ± 3.8	0.0–19.0
Father’s education (years of schooling)	7.7 ± 4.5	0.0–19.0
Household members (number)	4.9 ± 1.9	2–17
Siblings (number)	1.8 ± 0.9	0–6
Household income (USD/month)	320.7 ± 261.6	58.7–2937.7
Length of hospital stay (days)	5.4 ± 2.4	1.0–16.0
Average temperature (°C)	26.6 ± 3.6	14.8–32.5
Temperature variation (°C)	1.0 ± 0.6	0.0–3.4
Daily temperature range (°C)	2.5 ± 1.6	0.0–8.5
Average humidity (%)	73.0 ± 9.6	43.0–94.0
Humidity variation (%)	5.4 ± 2.9	0.0–20.1
Daily humidity range (%)	13.1 ± 8.1	0.0–47.0
Average rainfall (mm)	5.8 ± 10.8	0.0–149.0
Rainfall variation (mm)	7.6 ± 11.8	0.0–72.5
Daily rainfall range (mm)	17.7 ± 28.6	0.0–149.0

SD = standard deviation.

**Table 3 children-08-01036-t003:** Bivariate and multivariate coefficients (β) with 95% Cl of explanatory variables by generalized linear model with gamma distribution.

Variables	Bivariate Log-Link β^2^ (95% CI)	*p*-Value	Multivariate Log-Link β^2^ (95% CI)	*p*-Value
Age of the children	0.996 (0.994–0.999)	0.011	0.996 (0.994–0.999)	0.006
Sex of the children (male)	0.995 (0.936–1.057)	0.871	0.983 (0.932–1.036)	0.516
Maternal age (year)	1.000 (0.994–1.006)	0.981	0.996 (0.988–1.005)	0.390
Paternal age (year)	1.001 (0.997–1.006)	0.668	1.005 (0.999–1.011)	0.104
Mother’s education, year of schooling	0.996 (0.989–1.004)	0.311	0.995 (0.987–1.003)	0.234
Father’s education, year of schooling	0.999 (0.993–1.006)	0.850	1.003 (0.996–1.010)	0.431
Mother’s occupation				
Unemployment	0.908 (0.815–1.009)	0.058	0.947 (0.860–1.041)	0.266
Employment	Ref.		Ref.	
Father’s occupation				
Unemployment	1.278 (1.016–1.636)	0.002	1.219 (0.997–1.509)	0.060
Unskilled occupation	1.001 (0.942–1.062)	0.984	0.991 (0.936–1.048)	0.751
Skilled occupation	Ref.		Ref.	
Number of family member	1.020 (1.005–1.036)	0.018	1.020 (1.005–1.036)	0.010
Number of siblings	1.016 (0.983–1.050)	0.336	0.997 (0.961–1.035)	0.890
Household income	1.000 (1.000–1.000)	0.507	1.000 (1.000–1.000)	0.569
Nutritional status				
Well-nutrition (≥−2 ZWA)	0.925 (0.865–0.989)	0.024	0.936 (0.881–0.994)	0.031
Malnutrition (<−2 ZWA)	Ref.		Ref.	
Birth history				
Term baby	0.883 (0.714–1.078)	0.181	0.921 (0.771–1.090)	0.351
Preterm	0.909 (0.720–1.135)	0.356	0.932 (0.766–1.125)	0.469
Post-term	Ref.		Ref.	
Delivery				
Normal vaginal	0.983 (0.928–1.041)	0.551	1.007 (0.958–1.058)	0.787
Caesarean section	Ref.		Ref.	
Average temperature	0.997 (0.989–1.005)	0.515	1.001 (0.993–1.009)	0.742
Temperature variation	1.159 (1.105–1.217)	<0.001	1.047 (0.995–1.103)	0.079
Average humidity	0.999 (0.996–1.002)	0.544	1.003 (1.000–1.007)	0.081
Humidity variation	1.047 (1.037–1.057)	<0.001	1.040 (1.029–1.052)	<0.001
Average rainfall	0.997 (0.994–1.000)	0.161	0.980 (0.973–0.987)	<0.001
Rainfall variation	1.002 (1.000–1.005)	0.047	1.014 (1.008–1.019)	<0.001

β = beta, Cl = confidence interval, *p*-value < 0.05.

## Data Availability

The data presented in this study are available on request from the corresponding author. The data are not publicly available due to research participants’ privacy.

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
