# Peer review of "Association of Socio-Demographic and Climatic Factors with the Duration of Hospital Stay of Under-Five Children with Severe Pneumonia in Urban Bangladesh: An Observational Study"

_children, 2021, doi:10.3390/children8111036_

Round 1

Reviewer 1 Report

In this manuscript, the authors have examined associations between a range of socio-demographic and climate variables with the length of hospital stays of 944 children aged <5 years with severe pneumonia, who were admitted to hospitals located within urban areas of Bangladesh. Using a generalized linear model with gamma distribution to analyze socio-demographic background, birth details, delivery history and nutritional status of the children, as well as local weather data, significant positive associations were observed between length of stay (LOS) and the number of household members and variations in humidity and rainfall.

The study adds to the important and growing body of studies examining the effects of climate change on children’s health and it is highly relevant that this research has been conducted in Bangladesh, a country which is greatly impacted by global warming and which lacks resources to universally provide air conditioning and other forms of indoor climate control, especially within hospitals. Studies attempting to examine the direct effects of climate change on health face significant challenges in gathering this evidence due to the influences of a multitude of confounding variables. By examining only children in this study with severe pneumonia who were all admitted to urban hospitals and diagnosed and managed by qualified physicians in the same manner, according to WHO guidelines, as well as carefully excluding children who were extremely malnourished, the authors have systematically attempted to reduce at least some of the effects of confounding factors on a single outcome variable (LOS), which is a marker for severity, which is a clear strength of the study. The authors have also been careful not to overstate the positive associations observed, especially with variations in humidity and rainfall as more research will clearly be required in order to substantiate such findings.

However, I would recommend the following minor amendments, which I feel would improve the overall quality of the manuscript:

i) In section 2.5 in the Methods (line 164), the addition of a statement to say that demographic categorical data were presented as frequencies would be helpful.

ii) In section 3.1 in the Results (line 202), the addition of the word “average” to the LOS results would be helpful to readers.

iii) Table 1.2 would be better titled as “Mean (SD) and range of other continuous demographic variables”.

iv) Adding the R-squared and p values to each of the graphs (A-E) in Figure 3 would enable the significant and non-significant associations between LOS and climate variables to be better visualized.

v) Please clarify why average rainfall in Table 2 was not considered to be significantly associated with LOS given that the p-value was significant (p<0.001).

vi) In section 4.1 of the Discussion, please clarify lines 359-361, starting with “Fourth, as we use weather station data,…..”. Do you mean to say in this sentence that as the weather variables of humidity and rainfall that you used in your analysis were not measured in the actual hospital environment, the effects on LOS need to be interpreted with caution? I would suggest rewording this sentence to improve clarity.

vii) In the Discussion, speculate on how increased variations in humidity and rainfall might affect LOS and delay recovery from severe pneumonia in young children.

viii) I feel that the Conclusion section is too long and unfocussed and would suggest revising. Apart from the summary of the findings at the beginning and the mitigation strategies at the end of this section, I would suggest including some points about the possible implications of these findings as well as some future directions for this research, e.g. could these be conducting prospective studies of hospitalized children with severe pneumonia where actual humidity and rainfall measurements are taken inside hospitals/within hospital grounds and perhaps limiting other confounding effects such as comorbidities?

Author Response

Date:   07 November, 2021
Subject: Re-submission of the [Children] Manuscript ID: children- 1442773

Thank you very much again for your valuable comments and further consideration of my manuscript for possible acceptance.

Please find below the point-by-point responses, and all the changes are marked by “Track Changes” within the revised manuscript.

Best regards,

KATM Ehsanul Huq
Doctoral student
Health Sciences Major
Graduate School of Biomedical and Health Sciences
Hiroshima University, Japan
Mobile:080 6266 8578

 Review Report Form 1:

Comment i) In section 2.5 in the Methods (line 164), the addition of a statement to say that demographic categorical data were presented as frequencies would be helpful.

Response: Thank you very much for your excellent comment. We have added the statement in page 4, line 165-166.

Comment ii) In section 3.1 in the Results (line 202), the addition of the word “average” to the LOS results would be helpful to readers.

Response: Many thanks. We have added the word in the page 5, line 205.

Comment iii) Table 1.2 would be better titled as “Mean (SD) and range of other continuous demographic variables”.

Response: Thanks for your valuable suggestion. We have changed it accordingly in page 6, line 208.

Comment iv) Adding the R-squared and p values to each of the graphs (A-E) in Figure 3 would enable the significant and non-significant associations between LOS and climate variables to be better visualized.

Response: Thanks for your suggestion. We have added the values in the page 8-13.

Comment v) Please clarify why average rainfall in Table 2 was not considered to be significantly associated with LOS given that the p-value was significant (p<0.001).

Response: We thank you for the valuable comments. We are sorry for the wrong interpretation. We have corrected it in page 1, line 28 & 31; page 13, line 273-274, 278-279 & 281-284; page 14, line 296; page 15, line 334-336 & 342.

Comment vi) In section 4.1 of the Discussion, please clarify lines 359-361, starting with “Fourth, as we use weather station data,…..”. Do you mean to say in this sentence that as the weather variables of humidity and rainfall that you used in your analysis were not measured in the actual hospital environment, the effects on LOS need to be interpreted with caution? I would suggest rewording this sentence to improve clarity.

Response: Thanks for your core query. We have added your valuable suggestion in the page 16, line 383-386.

Comment vii) In the Discussion, speculate on how increased variations in humidity and rainfall might affect LOS and delay recovery from severe pneumonia in young children.

 Response: Thanks for your core query. Due to the lack of previous evidences, this is really difficult to speculate the potential reasons of association between the increased variations in humidity and rainfall with lesser average rainfall and increased LoS and the delay in recovery from severe pneumonia in young children. However, we have addressed this mitigation issue to shorten the LoS and prevent severe pneumonia recovery time.

Comment viii) I feel that the Conclusion section is too long and unfocussed and would suggest revising. Apart from the summary of the findings at the beginning and the mitigation strategies at the end of this section, I would suggest including some points about the possible implications of these findings as well as some future directions for this research, e.g. could these be conducting prospective studies of hospitalized children with severe pneumonia where actual humidity and rainfall measurements are taken inside hospitals/within hospital grounds and perhaps limiting other confounding effects such as comorbidities?

Response: We quite concur with your thoughtful suggestions. We have changed the ‘Conclusions’ section and added your valuable suggestions in the page 16-17, line 396-418.

Reviewer 2 Report

The study is interesting and relevant. The paper is clear and objective. The method design is solid. the statistical conclusion distinguish between correlational and causally determined variables. The external validity should be tested in some other countries with similar socio-demographic and climatic factors.

Author Response

Date:   07 November, 2021
Subject: Re-submission of the [Children] Manuscript ID: children- 1442773

Thank you very much again for your valuable comments and further consideration of my manuscript for possible acceptance.

Please find below the point-by-point responses and all the changes are marked by “Track Changes” within the revised manuscript.

Best regards,

KATM Ehsanul Huq
Doctoral student
Health Sciences Major
Graduate School of Biomedical and Health Sciences
Hiroshima University, Japan
Mobile:080 6266 8578

 Review Report Form 1:

Comment 1) The external validity should be tested in some other countries with similar socio-demographic and climatic factors.

Response: Thank you very much for the insightful comment. We added your valuable suggestion in the ‘Conclusions’ section of page 17, line 416-418.